# Notch Signaling in Insect Development: A Simple Pathway with Diverse Functions

**DOI:** 10.3390/ijms241814028

**Published:** 2023-09-13

**Authors:** Yao Chen, Haomiao Li, Tian-Ci Yi, Jie Shen, Junzheng Zhang

**Affiliations:** 1Department of Plant Biosecurity and MOA Key Laboratory of Surveillance and Management for Plant Quarantine Pests, College of Plant Protection, China Agricultural University, Beijing 100193, China; s20193192642@cau.edu.cn (Y.C.);; 2Guizhou Provincial Key Laboratory for Agricultural Pest Management of Mountainous Regions, Institute of Entomology, Guizhou University, Guiyang 550025, China

**Keywords:** Notch, insect, development, *Drosophila*

## Abstract

Notch signaling is an evolutionarily conserved pathway which functions between adjacent cells to establish their distinct identities. Despite operating in a simple mechanism, Notch signaling plays remarkably diverse roles in development to regulate cell fate determination, organ growth and tissue patterning. While initially discovered and characterized in the model insect *Drosophila melanogaster*, recent studies across various insect species have revealed the broad involvement of Notch signaling in shaping insect tissues. This review focuses on providing a comprehensive picture regarding the roles of the Notch pathway in insect development. The roles of Notch in the formation and patterning of the insect embryo, wing, leg, ovary and several specific structures, as well as in physiological responses, are summarized. These results are discussed within the developmental context, aiming to deepen our understanding of the diversified functions of the Notch signaling pathway in different insect species.

## 1. Introduction of the Notch Signaling Pathway

A small number of signaling pathways are iteratively used to regulate cell fate determination, organ growth and tissue patterning during insect development. One of these important pathways is mediated by Notch, which functions to distinguish adjacent cells in numerous developmental processes [1]. The Notch signaling cascade operates in a remarkably linear manner without the signal amplification steps which are normally found in other pathways [2]. The core components of the Notch signaling pathway include the ligands of Delta and Serrate (known as Jagged in vertebrates), the Notch receptor and the CBF1/Su(H)/LAG1 (CSL) family transcription factors [3]. Both the Notch and Delta/Serrate (Dl/Ser) proteins contain a large extracellular domain composed of epidermal growth factor-like repeats which are pivotal for their direct contact. Notch recognizes Dl/Ser presented at the surface of neighboring cells and the interaction with ligands triggers the proteolytic cleavage of the Notch protein [4]. This process releases the intracellular domain of Notch (NICD), which is subsequently translocated into the nucleus. Within the nucleus, NICD forms a transcription activation complex with Su(H) and the co-activator Mastermind (Mam) to drive the transcription of downstream target genes [5]. In the absence of signal input, NICD is not produced and Su(H) recruits the co-repressors CtBP and Hairless to suppress the expression of Notch targets. Thus, Notch detects signals sent by neighboring cells, transduces these signals and adjusts the cell state accordingly (Figure 1).

Notch signaling is conserved across the animal kingdom [6,7,8]. Notch signaling plays crucial roles in various developmental events, encompassing cell fate determination, cell cycle progression, cell death and stem cell homeostasis [1,2,3,4,5]. Therefore, it is not surprising that mutations of genes encoding Notch signaling components contribute to various human diseases, including birth defects and malignant tumors [9,10,11,12]. Many aspects of the Notch signaling pathway, such as the signal transduction machinery, the regulatory network, as well as the involvement in human diseases, have been elaborately reviewed in previous articles [13,14,15,16,17,18,19,20]. This review will focus on the diversified roles of Notch signaling in insect development, covering recent findings in various insect species.

## 2. A Brief Historical Review of Notch Signaling Studies in *Drosophila*

More than a century ago, Thomas H. Morgan chose *Drosophila melanogaster*, the fruit fly, as a model organism to study the fundamental law of heredity. The discovery of the famous white eyed mutant fly led to a series of significant advances toward the modern theory of genetics [21,22]. Alongside the eye color mutants, Morgan and his students isolated and characterized various mutants affecting body color and wing morphology [23]. In 1911, Morgan reported the generation of “beaded wings” mutants by radium ray irradiation, and recorded that the marginal vein and wing blade in these mutants were eliminated to various degrees [24]. In 1914, John S. Dexter isolated one mutant strain from the beaded stocks which he named as “Perfect Notched”, and demonstrated that a dominant sex-linked factor was responsible for the wing notches in the tips [25]. Regrettably, the “Perfect Notched” stock was lost and Dexter was not able to further examine the factor underlying the notched wing phenotype. Morgan and his students identified additional *Notch* mutants and Calvin B. Bridges mapped the approximate position of *Notch* in the X chromosome [23]. In 1919, Otto L. Mohr recovered a novel *Notch* allele and showed for the first time that the deficiency of a small region of the X chromosome likely caused the notched wing phenotype [26]. In the next seven decades, the number of *Notch* alleles and related phenotypes continued to grow, while the biochemical nature of the Notch protein remained a mystery [27]. In 1983, the *Notch* gene was cloned and the finding that the *Notch* gene encodes a transmembrane receptor inspired subsequent studies regarding cell–cell interaction and signal transduction [28,29,30,31]. Thereafter, detailed molecular genetic analyses have defined the core components and the canonical signal transduction cascade of the Notch signaling pathway.

Mutations of Notch pathway components are found to affect a wide array of developmental events, including cell differentiation, tissue pattern formation and stem cell self-renew [32,33,34]. It is now widely acknowledged that Notch signaling plays a pivotal role in the development of most, if not all, organs in the fruit fly [35,36,37,38]. Three primary modes of Notch actions have been discovered: lateral inhibition, boundary induction and lineage decision [32]. An early reference to the concept of “lateral inhibition” is found in the study of cuticle patterning in the bloodsucking bug *Rhodnius prolixus* [39]. In the dorsal abdomen of *R. prolixus*, bristles arise from small mounds of smooth cuticle termed plaques, and each plaque appears to exert an inhibitory effect that prevents the formation of new plaques within a certain radius [39]. Lateral inhibition operates within a cluster of cells initially sharing a similar fate and potential, whereas Notch signaling amplifies minor differences among them. Consequently, a cell displaying low or no Notch activity acquires a specific fate and prevents the surrounding cells from adopting the same developmental route [32]. When acting between two cell populations, Notch signaling establishes a boundary to segregate the two cell groups, a process commonly employed to subdivide developmental fields during tissue formation. Through the asymmetrical distribution of signaling regulators which leads to differential signaling activity, Notch controls the binary specification of cell fate between daughter cells in various cell lineages [32]. It has become evident that Notch functions at different developmental stages and within multiple tissues in the fruit fly, sometimes employing distinct modes of action within a single tissue. Systematic in vivo screens have identified extensive sets of genes that are involved in Notch signaling within diverse developmental contexts [40,41,42,43,44,45]. The exploration of the molecular functions of these genes will shed light on how Notch signaling accomplishes such sophisticated roles.

The diversified developmental functions of the Notch signal have been explored in depth in the model insect *D. melanogaster*. Recent advances in genomic resources and genetic tools have allowed investigations of the Notch function among a broad range of insect species, which will be discussed in the following sections.

## 3. Notch Signaling in Insect Embryonic Neurogenesis

Donald F. Poulson is generally regarded as the pioneer establishing the connection between Notch and insect embryo development. In the 1930s, Poulson described a unique “neurogenic” phenotype in *Notch* mutant embryos of *D. melanogaster*. These embryos failed to develop mesodermal and endodermal tissue while concurrently exhibiting an overgrowth of the nervous system [46,47]. Subsequent genetic screens discovered that mutations of core Notch signaling components *Dl*, *mastermind* (*mam*) and *Enhancer of Split* (*E*(*spl*)), led to similar neurogenic defects [48,49]. Notch signaling acts at multiple stages to control embryonic nervous system development, including a selection of neural progenitor cells (neuroblasts; NBs), control of NBs daughter cells proliferation, specification of neuronal cell fate, glia development and axon pathfinding [50]. The fly embryonic NBs are selected from a uniform layer of cells, referred to as neuroectoderm, with lateral inhibition playing a pivotal role [51]. Expression of proneural genes defines stereotypically spaced “proneural clusters” in the neuroectodermal cell sheet, each proneural cluster consists of 6–8 cells with similar potential to develop as NBs [52]. Within these proneural clusters, cellular interactions mediated by a Notch signal culminating in the selection of the cell with the lowest Notch activity to become NB. The activation of Notch signaling in cells surrounding the NB results in the expression of transcription factors encoded by the *E*(*spl*) gene complex. The E(spl) proteins directly repress proneural genes, effectively preventing these cells from adopting a neuroblast fate [52]. In mutants of *Notch* and many other Notch pathway genes, excessive NBs are formed due to the lack of lateral inhibition within the proneural clusters [53].

NBs were recognized as a distinct population of cells with specific characteristics of cell size, cell shape, and nuclear position in the embryos of various insect species more than 130 years ago [54]. A similar pattern of NBs emerged across insects with different developmental modes and life histories, including *D. melanogaster*, the bloodsucking bug *R. prolixus*, cockroaches (*Blatta germanica* and *Periplaneta americana*), locusts (*Locusta migratoria*, *Melanoplus femurrubrum*, *Schistocerca americana* and *Schistocerca gregaria*), potato beetle (*Leptinotarsa decemlineata*), red flour beetle (*Tribolium castaneum*), stick insect (*Carausius morosus*), silverfish (*Ctenolepisma longicaudata*), tobacco hornworm (*Manduca sexta*) and yellow mealworm (*Tenebrio molitor*) [54,55,56,57,58,59,60,61,62,63,64,65,66]. These observations suggest the possibility of conserved mechanisms mediating the selection of NBs during insect embryo development [67,68,69]. The landmark laser ablation experiments conducted in the locust (*S. americana*) embryos demonstrated that the enlarging NB enforces lateral inhibition, ensuring the formation of just one NB within each proneural cluster [59,66]. In cricket (*Gryllus bimaculatus*) and cockroach (*P. americana*) embryos, knock-down of *Notch* and/or *Dl* by RNA interference (RNAi) caused a classic neurogenic phenotype in early stages and subsequent cell apoptosis in later stages [70,71,72]. RNAi knock-down of *Notch* and *E*(*spl*) led to neurogenic phenotypes with an elevated number of NBs in red flour beetle embryos [73]. Computational analyses suggest a notable conservation of the *E*(*spl*) gene family among insects [74,75,76,77,78,79,80]. It is possible that Notch signaling and the *E*(*spl*) gene family commonly contribute to determining NB fate in insect embryos [81].

## 4. Notch Signaling in Insect Embryo Segmentation

Beyond its ubiquitous role in neurogenesis, Notch signaling is also recognized to govern diverse embryonic developmental events in different insects. It is evident that Notch signaling is critical for embryo segmentation in several insect species, while being nonessential for this process in others [82]. Segmentation is a fundamental process that divides the developing body into separate units, each capable of undergoing independent developmental programs [83]. Insects exhibit two distinct modes of embryo segmentation [84]. In long germ insects such as *D. melanogaster*, all segments are specified nearly simultaneously within the blastoderm prior to gastrulation. Conversely, in insects with short and intermediate germ, only segments of the head region are specified in the blastoderm, with the remaining segments arising sequentially from a posterior segment addition zone using a clock and wave front mechanism akin to vertebrates [84]. Many segmentation factors originally identified from genetic screens in *D. melanogaster* exhibit conserved functions among insects with different germ types [82,83,84,85]. Notch signaling plays critical roles during embryo segmentation in vertebrates and several sequential segmenting arthropods, including brine shrimp, water fleas, centipedes and spiders [82]. It has been hypothesized that Notch signaling represents ancestral mechanisms governing segmentation in arthropods and vertebrates [86]. However, the extent and manner in which Notch signaling is implicated in insect embryo segmentation is still under debate.

It has been shown that Notch signaling is dispensable for embryo segmentation in *D. melanogaster*. Despite notable neurogenesis defects arising in later stages, the segment morphology and expression pattern of segmentation factors remained unaffected in *Notch* mutant fly embryos [87,88]. In another long germ band insect, the honeybee (*Apis mellifera*), Notch signaling was also not implicated in segmentation [89]. In the short germ band milkweed bug *Oncopeltus fasciatus*, the expression pattern of *Dl* was incongruous for regulating embryo segmentation [90,91]. Consistently, RNAi knock-down of *Dl* in an *O. fasciatus* embryo failed to affect the expression pattern of other segmentation factors [92]. Likewise, in the short germ band red flour beetle, there is no substantial evidence supporting the role of Notch signaling in segmentation [93]. The *fringe* (*fng*) gene, known to encode a conserved modifier of the Notch receptor [94,95], is essential for segmentation in mice and chicken [96,97]. Yet, in the short germ locust *S. gregaria*, *fng* expression becomes detectable only after segment boundaries are established, thereby excluding its involvement in embryo segmentation [98]. The germ band morphology is highly dynamic in the silkworm *Bombyx mori*, yet the available molecular data indicate that the majority of segments are not patterned prior to gastrulation, aligning with the short germ type [84]. In *B. mori* embryos, *Notch* RNAi caused patterning defects without affecting the formation of segments [99]. Conversely, *Dl* RNAi led to a loss of posterior segments and a disruption of segment boundaries in *B. mori* [100]. The cockroach *P. americana*, classified as a short germ type, exhibited segment morphology defects and alterations in the expression pattern of segmentation factors upon *Notch* RNAi [72,86]. In the intermediate germ cricket *G. bimaculatus*, Notch signaling was maternally required for morphogenesis of embryo segments and formation of posterior segments [70]. However, a subsequent study contested the necessity of zygotic Notch signaling for the establishment of segment boundaries. The authors argued that apoptosis and neurogenesis defects during early stages might lead to secondary effects in segment morphologies [71]. In conclusion, substantial evolutionary flexibility exists among the insects regarding how to divide segments in the embryo. There is no definitive correlation between the germ type and the involvement of Notch signaling in embryo segmentation.

## 5. Notch Signaling in Insect Wing Development and Patterning

Insects stand as the sole group of invertebrates to possess wings, a key evolutionary innovation that propelled them to the forefront of diversity and abundance within the animal kingdom [101]. While the evolutionary origin of insect wings remains a debated enigma, the fundamental steps and signaling pathways underlying wing development are quite conserved among winged insects [101,102,103]. Insights of how a Notch signal regulates insect wing development largely come from studies in *D. melanogaster*. In fruit fly wings, Notch signaling regulates various developmental events, including wing margin formation, wing growth, vein patterning and sensory organ specification [104,105].

The *Notch* gene is named after the phenotype of “one or more incisions at the end of wings”, which is arguably the most common and prominent defect observed in *Notch* mutant flies [22]. Yet, it took more than a century to unravel the cellular and molecular mechanisms by which Notch signaling regulates various aspects of fly wing development [23,104]. As a typical holometabolous insect, the fruit fly undergoes complete metamorphosis, implying that the larvae bear no resemblance to the adult and the transformation to adult occurs during the pupal stage. The precursors of adult wing persist as distinct clusters of undifferentiated cells called the wing imaginal disc (also known as wing disc) in the larval stages [105]. Despite the significant difference in cell number, cell size, cell identity and tissue morphology between the wing disc and adult wing, most of the wing patterning events take place in the wing discs [104]. In the developing wing disc, cells utilize Notch signaling to establish the boundary between the dorsal and ventral (D/V) compartment. In the adult wing blade, cells in these compartments emerge as the two apposed epithelial sheets, while the D/V boundary cells form the wing margin [105]. Notch activation occurs at both sides of the D/V boundary, facilitated by two different ligands: Dl activates Notch in dorsal boundary cells and Ser activates Notch in ventral boundary cells [106,107,108]. Glycosylation in the extracellular domain by Fng imparts Notch with an affinity for binding with Dl, while inhibiting its binding with Ser [109,110,111,112,113,114,115,116]. The expression of both *Ser* and *fng* is controlled by the dorsal-specific transcription factor Apterous (Ap), and feedback loops among these genes further strengthen the D/V boundary [110,117,118,119,120]. In the D/V boundary cells, Notch signaling promotes their proliferation and survival through activating the expression of target genes such as *vestigial* (*vg*), *wingless* (*wg*) and *cut* [106,107,121,122,123,124,125,126,127,128,129,130,131]. Beyond their cell autonomous functions, Vg and Wg also regulate the growth of cells distanced from the D/V boundary [106,121,128,129]. Mutations impairing Notch signal activity disrupt the segregation of the D/V compartment as well as the overall growth in the wing [105].

The fly wing blade consists of two main cell types: vein and intervein. Veins serve as structural supports for the wing blade and as vessels for trachea, nerves and hemolymph [104]. Notch signaling promotes intervein fate and inhibits vein fate, thereby establishing the boundary between the two types of cells [132,133,134,135]. Mutations dampening Notch signal activity during both the larval and pupal stages yield veins with uniformly increased thickness and deltas at their tips. Conversely, the inappropriate activation of Notch signaling can lead to the loss of adult veins [118,132,133,134,136]. The wing disc contains a small number of sensory organ precursor (SOP) cells which will form sensory bristles in the notum and along the anterior edge of the wing margin in adult flies [105]. The selection of SOPs is governed by a lateral inhibition process orchestrated by Notch signaling. SOPs undergo stereotyped asymmetric divisions to form the mechanosensory organ and the fates of daughter cells are also regulated by Notch signaling [37,122,137,138,139]. A disruption of Notch activity at different developmental stages could disrupt bristle pattern and bristle number, as well as cell lineage specification [1,139]. The aberrations in wing margin, veins and sensory bristles have become easily recognizable and reliable indicators used in genetic screens that aim to identify Notch signaling modulators [140,141,142].

Notch signaling is required for wing development in several Dipterans. In fly species closely related with *D. melanogaster*, such as *Drosophila hydei* and *Drosophila virilis*, mutant alleles of *Notch* and other genes in the pathway led to similar wing margin, vein and bristle defects [143,144,145,146,147,148,149,150]. Mutations displaying nicked wing margins have been isolated in the housefly *Musca domestica*, which were later mapped as *Notch* and *cut* mutant alleles [151,152,153]. With the completion of *M. domestica* genome sequencing and the success of Cas9-mediated genome editing, further molecular genetics analyses will provide insights about the roles of Notch signaling in house fly development [154,155]. Many mutations affecting wing development have been isolated and characterized in the Australian sheep blowfly, *Lucilia cuprina* [156,157,158]. The *Scalloped wings* (*Scl*) loss-of-function mutants displayed wing notching, vein thickening and bristle abnormalities as well as an embryo neurogenesis defect, and the *Scl* gene has been molecularly identified as the homolog of *Notch* [159,160].

Lepidoptera insects such as butterflies and moths normally possess two pair of wings (forewings and hindwings) covered by microscopic dust-like scales. Although the wing structure and morphology markedly differ from that of the flies, a series of works have underscored the importance of Notch signaling during *B. mori* wing development. The *flügellos* (*fl*) mutant of the silkworm produce wingless pupae and moths due to an inability of wing discs to respond to ecdysone during metamorphosis. The *fl* mutant wing discs develop normally until the fourth larval instar, with the defects manifesting in the fifth larval instar and pupae [161,162,163,164]. Molecular mapping and cloning have affirmed that the *fl* locus houses the homolog of the *fng* gene [165]. A whole-mount in situ hybridization in wing discs revealed a dorsal layer-specific expression pattern of *fng* and the dorsal-ventral boundary expression of *BmWnt1*, the homolog of *wg* [165,166]. In *fl/fng* mutant wing discs, the expression of *BmWnt1* was diminished [165]. The microRNA *mir-2* was found to target *fng*, with an over-expression of *mir-2* and somatic mutagenesis of *fng* using the CRISPR/Cas9 system, resulting in similar wing morphology defects and *BmWnt1* expression inhibition [167]. Similar as observed in the fruit fly, the selector gene *ap* is expressed in the dorsal layer of the wing disc and *vg* mutants exhibit small wing phenotypes [168]. These findings indicate that *B. mori* likely employs the same regulatory module, including the compartment selector (Ap), Notch signaling cascade and downstream targets to govern D/V boundary formation and wing growth.

Butterflies are renowned for their colorful wings with many types of pattern elements and colors. In *Precis coenia*, the buckeye butterfly, *ap* expression was confined to dorsal cells while *wg* was expressed in cells along the future wing margin in the fifth instar wing discs [169]. This expression pattern resembles that of the fruit fly and silkworm [165,169]. Studies in multiple butterfly species proposed that *Notch* up-regulation likely represents an early step in eyespots and the formation of other wing color patterns [170,171,172,173,174,175,176,177]. The expression pattern of the Notch protein in the pupal wing of *Heliconius erato* indicates that Notch-mediated lateral inhibition might underly butterfly wing scale organization [178]. However, functional studies are required to demonstrate the precise roles of Notch signaling during butterfly wing development.

The highly specialized forewings, called elytra, are considered as an important trait driving the successful radiation of Coleoptera insects (beetles) [179]. Differing from the hindwings, beetle elytra consist of thick, hardened and pigmented cuticles and many morphologically distinct features. Nevertheless, the expression pattern of key regulatory genes in the elytra and hindwings of *T. castaneum* was similar [180]. The expression of Notch target genes, *wg* and *cut*, was found in D/V boundary cells in both elytra and hindwing discs [180]. RNAi knock-down of Notch pathway genes (*vg*, *nub* and *ap*) led to a wing growth defect and wing margin truncation [180]. The *abrupt* gene was found to encode a novel regulator of Notch signaling essential for wing vein patterning in both the red flour beetle and fruit fly [181]. Upon ectopic expression in the fly wing disc, E(spl) proteins from the red flour beetle suppressed the formation of bristles and veins [73]. The red flour beetle has emerged as an important model insect with elaborate genetic toolkit [182,183]. In the foreseeable future, studies in *T. castaneum* will bring up new insights about the roles of Notch signaling across different insect wing types.

A recent study in the brown planthopper, *Nilaparvata lugens*, revealed that the planthopper *Notch* gene encodes multiple protein isoforms by alternative splicing [184]. When dsRNA targeting different isoforms were injected into the planthopper nymphs, several Notch variants were found to regulate bristle, vein and wing blade development [184]. While the expression of Notch pathway genes has been detected in the wing discs of various insect species, their specific functions require further investigation [185,186].

## 6. Notch Signaling in Insect Leg Development

Notch signaling plays diverse and fundamental roles in leg patterning and growth in *D. melanogaster* [187]. The legs of the fruit fly are composed of ten segments, each separated by a flexible joint. A fusion of leg segments and a reduction in leg growth have been noticed in *Notch*, *Dl* and *Ser* mutants [188,189,190]. On the other hand, an ectopic activation of Notch signaling within the leg was sufficient to induce the formation of extra segment borders (joints) and cell growth [191,192,193]. Segment separation in the larval stage is crucial for proper leg development. *Notch*, *Ser*, *Dl* and *fringe* are expressed in a segmentally repeated pattern in the imaginal leg disc [191,192,193,194]. Notch is asymmetrically activated at the distal side of the *Ser-* and *Dl*-expressing domains, forming nine rings along the proximal-distal axis in the leg disc [195]. A continual activation of Notch across segments leads to shortened legs with segmentation defects [191,192,193]. Many factors and pathways interact with Notch signaling to establish the segment boundaries, although the underlying mechanisms are not fully elucidated [196,197,198,199,200,201,202,203,204,205,206,207,208,209]. Proper Notch pathway activation is also vital for joint formation and leg growth [210]. The leg joints can be classified into the proximal “true joints” and the distal “tarsal joints”. True joint morphology varies, while tarsal joints consist of a proximal “socket” and a distal interlocking “ball” [195]. Both types of joints are shaped by Notch signaling, with distinct target genes activated in true joints and tarsal joints [187,195,211]. Notch signaling is essential for the fate specification, cell shape changes and cell movements necessary for tarsal joint morphogenesis [212,213]. How Notch controls leg growth is largely unknown, with indications that an interaction with the Hippo pathway is involved [192,197].

The roles of Notch in leg segmentation, joint formation and leg growth are conserved across insect species. In *D. hydei*, *Notch* mutants exhibited tarsal elements fusion [146]. In the cricket *G. bimaculatus*, RNAi knock-down of *Notch* led to a reduced leg length and loss of joints [70]. The red flour beetle *T. castaneum* experienced joint loss but not leg length reduction after *Notch* RNAi, while *Ser* RNAi eliminated joints and reduced overall leg length [214,215]. The *nubbin* (*nub*) gene was expressed in a series of concentric rings in fly leg discs and a mutation of *nub* resulted in shortened legs [216,217]. Notch signaling directly regulates *nub* expression in fly leg discs [192,203]. The *nub* homologs were expressed in the developing legs in several insect species, including the cockroach *P. americana*, the milkweed bug *O. fasciatus* and the primitively wingless firebrat *Thermobia domestica* [218]. RNAi knock-down of *nub* in *O. fasciatus* embryos led to shortened thoracic legs and the growth of ectopic appendages on abdominal segments [219]. In the house cricket *Acheta domesticus* and the cockroach *P. americana*, *nub* was required for leg segment growth and joint formation [220]. In *P. americana,* the *nub* expression level was reduced after *Notch* RNAi, suggesting a potential regulatory role of Notch signaling on *nub* expression [220]. Several signaling pathways, including Notch, were upregulated during regenerative patterning and growth in ladybird beetle (*Harmonia axyridis*) legs [221]. Notch-mediated appendage segmentation has been proposed as an arthropod defining trait, which could be further tested in other insect species [222].

## 7. Notch Signaling in Insect Reproduction

Notch signaling plays essential roles during ovary development in *D. melanogaster*, particularly for egg chamber formation and the assembly and maintenance of the ovarian germline stem cell (GSC) niche [33,223,224].

Oogenesis in fruit flies initiates within the germarium located at the anterior tip of the ovariole. In the germarium, GSCs undergo asymmetric division to produce cystoblasts. Each cystoblast undergoes four rounds of incomplete cell division to generate a germline cyst containing 16 interconnected cells. Somatic follicle cells encapsulate the germline cyst; the collection of germline and follicle cells at this point is known as an “egg chamber.” Within the egg chamber, one germline cyst cell becomes the oocyte while the other cyst cells become nurse cells that contribute RNAs and proteins to the oocyte. The egg chamber progresses through numerous developmental stages and moves toward the posterior of the ovary before becoming a mature egg [224]. Temperature-sensitive mutant alleles of *Notch* and *Dl* significantly reduced the number of eggs laid by female flies. These mutations resulted in defects in follicle cell development and oocyte anterior-posterior (A-P) polarity [225]. Subsequent studies showed that Notch activity in follicle cells is essential for their transition from mitosis to endocycling, a process regulated by Dl expressed in germline cells [226,227,228,229,230,231,232,233,234,235,236,237,238]. The egg chamber possesses intrinsic A-P polarity, with nurse cells at the anterior and oocyte at the posterior [239]. Interestingly, this A-P polarity emerges through a relay mechanism that propagates asymmetry from older cysts to younger cysts. During early oogenesis, a germline cyst signals through the Dl-Notch pathway to induce the formation of anterior polar cells as it buds from the germarium. The anterior polar cells express the JAK/STAT signaling ligand unpaired, prompting the adjacent anterior stalk/polar precursor cells to adapt the stalk cell fate. The stalk guides the positioning of the oocyte at the posterior pole of the neighboring younger cyst through adhesive interactions. As oocyte positioning takes place, a new round of Dl-Notch signaling in the younger cyst induces anterior polar cells [240,241]. Thus, each cyst imparts polarity to the next cyst through a series of posterior to anterior induction events [242,243].

The concept of stem cell niche was proposed around half a century ago to describe the tissue microenvironment supporting the self-renew and maturation of haemopoietic stem cells, later extended to other stem cell-containing tissues. However, the ability of the niche to support stemness was first demonstrated in studies of fly ovarian GSC [244]. At the germarium’s anterior tip, the terminal filament cells and cap cells form the niche that sustains two to three ovarian GSCs. Proper Notch signaling is critical for the development of both terminal filament cells and cap cells [244,245,246,247,248,249,250,251,252,253]. The strength of Notch signaling also dictates the size of the ovarian GSC niche. The hyperactivation of Notch signaling yields more cap cells and larger niches, supporting more GSCs. Conversely, reduced Notch signaling results in decreased cap cell numbers and niche size, and less GSCs [246]. In mature adult flies, Notch signaling also conveys the impact of diet and age on GSC niche activity and GSC maintenance [254,255,256,257,258].

Two types of ovarioles, panoistic ovarioles and meroistic ovarioles, are present in insects. In panoistic ovarioles, all progenies of germline stem cell become oocytes and nurse cells are absent. Meroistic ovarioles contain nurse cells and can be further classified into polytrophic ovarioles and telotrophic ovarioles. In polytrophic ovarioles, nurse cells are found within the egg chamber and transport mRNA and proteins to oocyte through ring canals. While in telotrophic ovarioles, nurse cells reside in the germarium and are connected to early stage oocytes by nutritive cords [259]. Although it is generally believed that panoistic ovaries represent the ancestral type from which meroistic types had derived, there is no precise correlation between ovariole type and phylogenetic position [260]. In the panoistic ovariole of the cockroach *Blattella germanica*, inhibiting Notch signaling caused defects in stalk formation, follicle cell proliferation and follicle cell differentiation [261,262,263,264]. In the telotrophic ovariole of *T. castaneum*, Notch signaling is vital for stalk formation, follicle cell proliferation and establishing an A-P axis [265,266]. Interestingly, the role of Notch signaling in follicle cells varies significantly across different insect species. In *D. melanogaster*, Notch promotes the switch from mitosis to endocycle in follicle cells, whereas in *B. germanica* and *T. castaneum,* Notch signaling is essential for maintaining the mitotic cycle. The connection between Notch’s role in follicle cells and ovariole structure remains an open question. Notch was found to regulate vitellogenesis in the *L. migratoria* fat body, a process critical for oocyte maturation and ovarian growth [267]. Vitellogenesis is generally required for insect oogenesis and egg production; whether Notch signaling plays a role in this basic physiological event in other insects could be further examined [268]. In the honeybee *A. mellifera*, Notch signaling represses oogenesis in the germarium of worker bees [269,270]. In honeybee queens, Notch pathway genes were dynamically expressed in the ovariole, but their functions have not been examined [271,272]. Abundant Notch-like proteins were identified in the early stage *Bactrocera dorsalis* ovary, indicating a potential role in oriental fruit fly oogenesis [273]. Further investigations will help us to understand how Notch carries out specific roles in distinct cell types, developmental stages and insect species during ovary development and reproduction.

## 8. Notch Signaling in Insect Physiological Activities

Recent studies have discovered that Notch signaling is involved in regulating insect physiological activities across various conditions. Nutrition is one of the most important environmental variables impacting insect life history. Insects adjust their physiological activities and metabolic programs in response to changes in food quality and quantity [274]. The insulin pathway functions as a nutrition sensor, orchestrating metabolic requirement and other biological events [268]. In female insects, reproduction requires a massive input of nutrition resources to produce eggs enriched with nutrient reserves [275]. The insulin pathway transmits the diet’s impact on reproduction via the regulation of Notch signaling in the *D. melanogaster* ovary [254,255,256,258,276]. In somatic tissues, such as the gut, muscle and neuronal system, insulin signaling influences Notch activity through diverse mechanisms [277,278,279]. A recent study found that dietary cholesterol influences the level and duration of Notch signaling by modulating Dl and Notch stability and trafficking, which in turn impacts cell differentiation in fly adult midgut and alters the metabolic program [280]. The expression of Notch pathway genes was upregulated upon provision of a high-quality diet to the honeybee *A. mellifera* [281]. In the larval guts of the Asian honey bee, *Apis cerana*, the Notch pathway appeared to be targeted by miRNAs and piRNAs at different developmental stages [282,283]. A supplementation of pterostilbene, fucoxanthin and a traditional Chinse herb *Cistanche tubulosa* extended the lifespan of fruit fly adults and increased the expression of Notch pathway genes [284,285,286]. These studies highlight the roles of Notch signaling in responding to nutrition status in insects.

Many viruses pose significant threats to human health and can be transmitted by vector insects such as mosquitoes [287]. Following infection with the alphavirus Sindbis, expression of Notch pathway genes increased in S2 cells, suggesting that Notch signaling may be involved in the establishment of virus persistence in insect cells [288]. The induction of Notch pathway genes was also observed upon Dengue virus infection in *Aedes albopictus* cells [289]. Infections with Dengue virus and Chikungunya virus led to up-regulation of Notch pathway genes and midgut cell division in the vector mosquito *Aedes aegypti* [290,291,292]. Knocking-down of *Dl* expression by RNAi inhibited the infection-induced midgut cell division, while significantly enhancing the susceptibility of the refractory *Aedes aegypti* strain to Dengue virus [293]. The human malaria parasite *Plasmodium berghei* also induced midgut cell division and activated the Notch pathway in the mosquito vector *Anopheles albimanus* [294,295]. In the midguts of *Anopheles gambiae*, *Plasmodium falciparum* infection induced changes in chromatin status within regulatory elements of Notch pathway genes, but the significance of these observations needs to be further explored [296]. In the larval midgut of the wild silk moth *Antheraea yamamai*, a pathogenic nucleopolyhedrovirus infection induced up-regulation of Notch pathway genes [297]. The inhibition of Notch signaling was associated with midgut development defects in locust *L. migratoria* and the yellow fever mosquito *Aedes aegypti* [298,299]. In the fruit fly midgut, Notch signaling drives asymmetric division in the intestine stem cells, governing tissue homeostasis and responses to various stimulations [300,301,302,303,304]. Whether virus and pathogen infection would trigger Notch-related responses in the intestinal tract could be further tested in other insects.

Notch plays crucial roles in the determination of hematopoietic cell fate and the maintenance of larvae lymph gland, a vital organ of the immune defense system in *D. melanogaster* [35,305,306,307,308]. In response to fungal infection and wasp parasitization, a reduction in Notch signaling activity triggers specific immune responses in fruit flies [308,309]. Gram-negative bacteria stimulation led to up-regulation of Notch pathway genes in honey bee workers [310]. Several lncRNAs were identified as regulators of immune priming in *T. castaneum*, likely acting through the modulation of Notch pathway gene expression [311]. These studies suggest that Notch signaling might be involved in specific immune responses upon pathogen infection in insects.

Using the developing wing as a model system, it has been observed that the anthrax toxins and cholera toxins inhibit endocytic trafficking of Notch signaling components and impair Notch activity in *D. melanogaster* [312,313]. In a Zika virus infection model, the non-structural virus protein NS4A was found to restrict fly eye growth through regulation of JAK/STAT signaling and to inhibit wing growth by affecting Notch activity [314]. Exposure to the heavy metal mercury resulted in neurogenesis defects in the embryo and marginal nicks in the wing, primarily through inhibiting NICD production in the fruit fly [315]. Treatment with methylmercury, an organic form of mercury easily absorbed by the intestinal trac and a common environmental pollutant, increased Notch signaling activity in fly cells and embryos [316,317,318,319]. The response of Notch target genes to mercury exhibited variations in different cell types, upon treatment with organic or inorganic forms and at times, independently of the Notch receptor [316,317,318,319]. Pesticide exposure also impacts Notch signaling in various insects. For instance, feeding fly larvae sublethal doses of chlorfenapyr resulted in developmental defects in the wing and leg and a disruption of Notch signaling activity [320]. Exposure to adverse environmental factors like a low dose of gamma-irradiation, formaldehyde, toluene and dioxin impaired Notch signaling in adult flies [321]. Harmine, a natural β-carboline, impaired fruit fly development by influencing Notch and other signaling pathways [322]. In the stingless bee *Partamona helleri*, fipronil exposure decreased Notch signaling activity in the brain and Malpighian tubules [323,324]. Although a sublethal level of fluralaner impaired larval development and led to wing notches in the common cutworm *Spodoptera litura*, the impact on Notch signaling remains unexamined [325]. Moreover, Notch signaling showed responsiveness to ultraviolet irradiation and metamorphosis oxidative stress in *B. mori* [326]. Collectively, these results suggest that Notch signaling may be involved in the response to hazardous factors in insects.

Insects can experience functional hypoxia when the oxygen supply is insufficient for metabolism demands and respond to hypoxia through diverse strategies [327]. In the fruit fly, Notch signaling regulates hypoxia tolerance, as flies with impaired Notch activity exhibit reduced hypoxia tolerance, whereas those with hyperactivated Notch signaling display the opposite effect [328,329,330,331,332]. It would be intriguing to explore whether such effects of Notch signaling in hypoxia tolerance are also present in other insects, particularly those experiencing environmental hypoxia at specific stages of their life cycle.

## 9. Notch Signaling in Less Studied but More Interesting Tissues

Insects have evolved to develop a multitude of captivating novel structures while keeping a steady basic body plan and the mechanisms responsible for the evolution of such morphological novelties remain puzzling [333]. From a developmental biology perspective, considering Notch signaling as one of the fundamental regulatory units of tissue development and growth, its involvement in shaping these morphological novelties is not surprising at all.

The bull-headed dung beetle *Onthophagus taurus* and numerous other scarab beetle species exhibit rigid projections of the exoskeleton from the thoracic and head regions referred to as horns. Beetle horns are highly diversified and have been viewed as an evolutionary novelty due to a lack of visible homology with existing structures. A previous study has found that beetle thoracic horns evolved from wing serial homologs [334]. Notch pathway genes were expressed in developing dung beetle horns [335,336]. Importantly, Notch signaling is a key regulator responsible for the dramatic diversity of male horn sizes and shapes within and across *Onthophagus* species [337]. In the Asian rhinoceros beetle *Trypoxylus dichotomus*, *Notch* RNAi disturbed horn primordial furrow depth, leading to defects in the horn shape [338].

The dorsomedial and the abdominal support structure are two types of body wall projections commonly observed in scarab beetle pupae. A study in *Onthophagus taurus* has revealed that these structures are indeed wing serial homologues and *Ser* RNAi disrupted the formation of both structures [339]. Another intriguing example is the “gin-trap”, a structure exclusively found on pupae of the closely related beetle families Tenebrionidae and Colydiidae. Gin-traps are believed to be evolved after the radiation of holometabolous insects and function as pupae defensive organs to grasp the appendages of predators. In the beetle *T. castaneum*, RNAi knock-down of Notch pathway components disrupted the formation of gin-traps [340].

In dung beetles, the fore tibia has transformed into a specialized digging tool that facilitates access to the compacted soil as a habitat. This fore tibia exhibits a flattened and enlarged configuration, possessing four to five prominent tibial teeth which enhance the digging performance. RNAi of *Ser* and downstream genes of Notch pathway resulted in a reduction in tibial teeth and a fusion of leg segments in *Onthophagus taurus* [341]. These findings underscore the recurrent utilization of Notch signaling in the development of evolutionarily novel morphological structures in beetles.

The insect antennae, serving as the principal olfactory sensory organs, are critical for locating food resources, finding mating partners, choosing oviposition sites, as well as for evading predators and toxic substances. The insect antennae exhibit remarkable diversity in shapes, structures and sizes [342]. In *D. melanogaster*, the antenna cell fate is determined by several selector genes, while Notch signaling regulates cell proliferation, tissue growth and the formation of boundaries between antenna segments [192,203,343,344,345]. In the beetle *T. castaneum*, *Ser* RNAi resulted in a strong reduction in antenna length and a complete absence of joints, whereas *Notch* RNAi led to the absence of antennal joints without significantly affecting antenna growth [346]. Interestingly, *Notch* RNAi, rather than *Ser* RNAi, decreased the density of sensory bristles on the antenna [346]. In the cricket *G. bimaculatus*, the *Notch* and *Dl* expression pattern suggested a potential role in antennal segmentation [70]. The *nub* gene, a down-stream target of Notch signaling during leg development in *D. melanogaster* and *P. americana*, also plays a role in antenna development [220]. In the milkweed bug *O. fasciatus*, *nub* RNAi resulted in sensory bristle defects without a significant impact on antenna segmentation and growth [219]. In *Acheta domesticus* and *P. americana*, a depletion of *nub* by RNAi led to the fusion of antenna segments [220]. Expression of the Notch target gene *E*(*spl*)*mβ* was detected in specific segments in *B. mori* larval antennal primordium, which develops into the feathery antenna seen in adults [347]. RNAi knock-down of *Notch* led to a significant fusion of antenna segments, an extensive reduction along the PD axis and milder defects such as lateral branch fusion in *B. mori* [348]. These observations support a general requirement of Notch signaling in insect’s antenna growth and segmentation.

Insects exhibit a remarkable diversity of mouthpart morphologies, yet the genetic regulatory network governing mouthpart development is not completely understood. RNAi knock-down of *Dl* in the silkworm and honeybee resulted in mild alterations in embryonic labrum shape [89,100]. Knock-down of two components of the Notch pathway, *Ser* and *mib1*, led to loss of the labrum in *T. castaneum* larvae presumably due to defects in cell proliferation [215]. RNAi of *Notch* and *Dl* also disrupted sensory organ development within the *T. castaneum* labrum [349]. The role of Notch signaling during mouthpart construction in the fruit fly and other insects has yet to be reported.

The diverse color patterns of insects often serve as camouflage to protect them from predators [350]. In the case of the Asian swallowtail butterfly, *Papilio xuthus*, young larvae exhibit black and brown patterns resembling bird droppings, transitioning to mimic host plants during their final instar. This change in color pattern is initiated by the juvenile hormone at the early fourth instar. Both *Dl* and *E*(*spl*)*mβ* were specifically expressed in the epidermis of the particular regions responsible for color markings during this transition phase [351,352]. A functional analysis demonstrated that Notch signaling defines the edge and pigmentation area of the final color patterns [353]. RNAi knock-down of *Notch* and *Dl* resulted in an expansion of the pigmentation area and a disruption of border lines in the fifth instar larvae. A similar but rather subtle change in larval color pattern was observed in *Papilio machaon*, a species closely related to *Papilio xuthus*, following *Dl* knock-down. In the silkworm *L* mutant larvae, which displays pairs of black brown twin spots on each body segment, knockdown of *Notch* but not *Dl*, *Ser* or *fringe* caused pigmentation loss in the twin spots [353]. These findings underscore the pivotal role of Notch signaling in the adaptive evolution of camouflage formation in caterpillars, motivating further exploration of the contributions of Notch signaling in color pattern development across diverse insect species.

## 10. Conclusions

Over a century has passed since the discovery of the first *Notch* mutant in the fruit fly, and this small insect has served as a prominent model system for dissecting the developmental roles of Notch signaling. Comprehensive studies in this little bug have yielded remarkable advancements in understanding the mechanisms of Notch signaling. The crucial components, the signal transduction cascade and the principal modes of action of the Notch pathway appear to be conserved across insect species. The participation of Notch signaling in the development of diverse insect tissues has been substantiated (Table 1).

Nevertheless, numerous fascinating developmental phenomena that are absent in the fruit fly exist across various insect species [58,84,342,350,354,355]. Recent advances in functional genetics tools such as genome editing in ‘non-model’ insect species have made it feasible to uncover novel factors and evaluate the roles of Notch signaling in diverse developmental phenomena. Future studies will undoubtedly help us to better understand the extensive roles of Notch in shaping insect tissues and could also reveal novel regulators, functions and signaling mechanisms. Importantly, these insights could be readily harnessed for the design of genetic control strategies such as RNA pesticides [356], genetic sexing [357] and gene drive systems [358], with the aim of safeguarding crops and humans against insect pests.

## Figures and Tables

**Figure 1 ijms-24-14028-f001:**
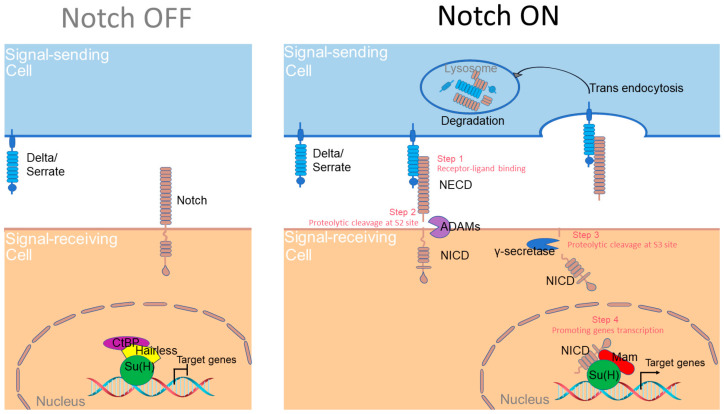
A simplified model of the Notch signaling transduction cascade in the fruit fly. The signal sending cell is in blue and the signal receiving cell is in earthy yellow. Upon binding with the Delta/Serrate ligands (Step 1), the Notch receptor sequentially processed metalloproteases of the ADAM/TACE family (Step 2) and the γ-secretase (Step 3), resulting in the release of NICD from the membrane. In the nucleus, NICD assembles with the CSL family protein Su(H) and the co-activator Mam to form a complex that regulates target gene expression (Step 4). In cells not receiving the activation signal, Notch is not processed and Su(H) interacts with co-repressors CtBP and Hairless to suppress the expression of target genes.

**Table 1 ijms-24-14028-t001:** Roles of Notch signaling across insect species.

Tissue/Organ	Biological Event	Species
Embryo	Neurogenesis	*Drosophila melanogaster* *Gryllus bimaculatus* *Periplaneta americana* *Gryllus bimaculatus* *Tribolium castaneum* *Lucilia cuprina*
	Embryo segmentation	*Bombyx mori* *Periplaneta americana* *Gryllus bimaculatus* ^a^
Wing	D/V boundary formation and wing margin integrity	*Drosophila melanogaster* *Drosophila hydei* *Drosophila virilis* *Musca domestica* *Lucilia cuprina* *Bombyx mori* *Tribolium castaneum* *Precis coenia* ^b^
	Wing growth	*Drosophila melanogaster* *Drosophila hydei* *Drosophila virilis* *Musca domestica* *Lucilia cuprina* *Bombyx mori* *Tribolium castaneum*
	Vein formation	*Drosophila melanogaster* *Drosophila hydei* *Drosophila virilis* *Musca domestica* *Lucilia cuprina* *Bombyx mori* *Tribolium castaneum* *Nilaparvata lugens*
	SOP selection and sensory bristle development	*Drosophila melanogaster* *Drosophila hydei* *Drosophila virilis* *Lucilia cuprina* *Nilaparvata lugens*
	Scale organization	*Heliconius erato* ^b^
Leg	Leg segmentation	*Drosophila melanogaster* *Drosophila hydei* *Gryllus bimaculatus* *Acheta domesticus* *Periplaneta americana* *Onthophagus taurus*
	Joint formation and morphogenesis	*Drosophila melanogaster* *Gryllus bimaculatus* *Tribolium castaneum* *Acheta domesticus* *Periplaneta americana* *Onthophagus taurus*
	Leg growth	*Drosophila melanogaster* *Gryllus bimaculatus* *Tribolium castaneum* *Oncopeltus fasciatus* *Acheta domesticus* *Periplaneta americana* *Onthophagus taurus*
	Leg regeneration	*Harmonia axyridis* ^b^
Ovary	Follicle cell differentiation and proliferation	*Drosophila melanogaster* *Blattella germanica* *Tribolium castaneum*
	Germline stem cell niche assembly and maintenance	*Drosophila melanogaster* *Apis mellifera* *Bactrocera dorsalis* ^b^
	Oocyte anterior-posterior polarity	*Drosophila melanogaster* *Tribolium castaneum*
	Vitellogenesis	*Locusta migratoria*
Physiological activity	Nutrition response in adult intestinal tract	*Drosophila melanogaster* *Apis mellifera* ^b^ *Apis cerana* ^b^
	Pathogen infection response in adult intestinal tract	*Aedes aegypti* *Aedes albopictus* ^b^ *Anopheles albimanus* ^b^ *Anopheles gambiae* ^b^ *Antheraea yamamai* ^b^
	Specific immune responses	*Drosophila melanogaster* *Apis mellifera* ^b^ *Tribolium castaneum* ^b^
	Mercury and pesticide toxicity	*Drosophila melanogaster* *Partamona helleri* ^b^ *Spodoptera litura* ^b^ *Bombyx mori* ^b^
	Hypoxia tolerance	*Drosophila melanogaster*
Other organs/tissues	Beetle horn development	*Onthophagus taurus* *Trypoxylus dichotomus*
	Dorsomedial and the abdominal support structure development	*Onthophagus taurus*
	Gin-trap development	*Tribolium castaneum*
	Digging tibia development	*Onthophagus taurus*
	Antenna growth and morphogenesis	*Drosophila melanogaster* *Tribolium castaneum* *Gryllus bimaculatus* *Oncopeltus fasciatus* *Acheta domesticus* *Periplaneta americana* *Bombyx mori*
	Mouthpart development	*Drosophila melanogaster* *Tribolium castaneum*
	Color pattern formation	*Papilio xuthus* *Papilio machaon* *Bombyx mori* Butterfly wing eyespots ^b^

^a^ Whether Notch signaling directly regulates cricket embryo segmentation is under debate. ^b^ Conclusion is based on gene expression pattern, functional studies are required.

## Data Availability

Not applicable.

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
