# Peer review of "Notch Signaling in Insect Development: A Simple Pathway with Diverse Functions"

_ijms, 2023, doi:10.3390/ijms241814028_

Round 1

Reviewer 1 Report

This review covers the topic of Notch signalling in insect development. It is challenging to write a comprehensive review on a topic such as Notch signalling, attempting to cover both the history of discoveries, findings in many insect species, and the current understanding of the molecular flow of the pathway. The review does a good job of the first two issues but is somewhat limited regarding the molecular details. But perhaps this one shortcoming is not an issue, given that the authors are not “advertising” the piece, in the Title and Abstract, as a detailed pathway review. I think this piece is well-written and will be of value for researchers studying insect species other than Drosophila melanogaster. I have only a few comments that could help improve the piece.

Figure 1: The yellow oval is labelled “co-activator”, but this entity is not described in the main text or figure legend.

Figure 1: The model is perhaps oversimplified, and e.g., the cleavage of Notch by ADAM proteases is generally viewed as occurring prior to ligand interaction. Moreover, Su(H) has functions, typically gene repression, in the absence of NICD.  

Some typos throughout e.g., “apterous (ap)” should be “Apterous (Ap)”, when referring to the factor, not gene.

The manuscript is well-organised but could benefit from a touch up regarding language.

Author Response

We thank the editor and the reviewers for critical reading of our manuscript and for their valuable suggestions. In the revised version of manuscript, we have made corrections, removed unnecessary figures and improved the quality of writing. We are confident that we have addressed the reviewers’ concerns in the current form of our manuscript. Our point-by-point response to the reviewers and editorial comments are listed as below and marked in red. Corrections made in the revised manuscript are also marked in red.

Reviewer 1:

This review covers the topic of Notch signalling in insect development. It is challenging to write a comprehensive review on a topic such as Notch signalling, attempting to cover both the history of discoveries, findings in many insect species, and the current understanding of the molecular flow of the pathway. The review does a good job of the first two issues but is somewhat limited regarding the molecular details. But perhaps this one shortcoming is not an issue, given that the authors are not “advertising” the piece, in the Title and Abstract, as a detailed pathway review. I think this piece is well-written and will be of value for researchers studying insect species other than Drosophila melanogaster. I have only a few comments that could help improve the piece.

Figure 1: The yellow oval is labelled “co-activator”, but this entity is not described in the main text or figure legend.

We thank the reviewer for pointing out that the yellow oval in the model is not described. The co-activator in the transcription complex is Mastermind (Mam). We have removed the yellow oval in the revised figure and included the description of Mam as the co-activator in the main text and figure legend.

Figure 1: The model is perhaps oversimplified, and e.g., the cleavage of Notch by ADAM proteases is generally viewed as occurring prior to ligand interaction. Moreover, Su(H) has functions, typically gene repression, in the absence of NICD.  

We appreciate the reviewer’s suggestions and have made corrections accordingly in the revised Figure. The cleavage of Notch by Furin at the S1 site indeed occurs prior to ligand interaction in the trans-Golgi network of Notch expressing cells (Blaumueller et al., 1997; Logeat et al., 1998). The cleavage of Notch by ADAM proteases at the S2 site requires ligand binding, and current model suggest that endocytosis of ligand bound to Notch likely generates a pulling force to induce substantial conformational movement that helps to expose the S2 site in Notch protein (Brou et al., 2000; Mumm et al., 2000; Wang et al., 2004; Tian et al., 2004; Gordon et al., 2007, 2015; Langridge and Struhl, 2017). The role of Su(H) in gene repression is illuminated in the revised figure.

Some typos throughout e.g., “apterous (ap)” should be “Apterous (Ap)”, when referring to the factor, not gene.

Changes have been made to ensure that the first letter of protein name is capitalized.

Comments on the Quality of English Language

The manuscript is well-organised but could benefit from a touch up regarding language.

We have thoroughly revised the manuscript to improve the language and modifications are marked in red.

Reviewer 2 Report

Report Notch review

The scholarly written review by Chen et al. reviews the literature about notch signaling in insects. Although numerous reviews have been published about Notch signaling in Drosophila, no such review is available that pretty systematically includes studies of Notch signaling in other insects in comparison to what we know from Drosophila. In every chapter of the review, the authors summarize the current understand of Notch signaling and then expand to other insects. Focusing in all chapters on molecular mechanisms, the review nicely complements and importantly extends existing entomology literature and especially entomology books. I would expect that the review will be found useful and informative for both entomologists finding out about molecular mechanisms or Drosophilists moving towards other insects. The list of references is long, and should remain like this, although I have not checked whether it comprehensively covers all related aspects. 

Beside a few minor point about language, I have a few suggestions which may improve the manuscript and its impact.

1. The overall structure of the article is fine with me. Chapter 3 “Notch signalign…” should be split into two “3. Neurogenesis” and “4. Embryonic Development”. The initial molecular experiments and mechanistic understanding of Notch were conducted concerning neurogenesis. Neurogenesis should therefore be visible in the structure of the article and should deserve a chapter on its own. The last chapter should be renamed, since “stress responses” are only a minor part of this chapter. A better title would be something like “adult physiology” or “homeostasis” or the like. 

2. Figures. I am not happy with the figures. Thinking which figure I could take for a lecture or a seminar, only figure 1 appears to be suitable. Figure 2 showing an evolutionary tree is currently without any information. The authors may compose a figure in which examples are shown of Notch signaling in species of the various branches. 

Figure 3 appears superficial. The authors may associate representative and the most striking examples cases to the three mechanisms for both Drosophila and other insects. I would recommend to stick to defined cases to avoid becoming superficial. 

In order to provide a comprehensive overview a table may be added. In a table it is much easier to provide a complete list. 

The article will have a much wider impact if accessible and useful/informative figures are included in the article.

3. Figure 1:  Please add numbers to the figure indicating the steps of the pathway. 1 on the left receptor-ligand binding, then 2 proteolytic cleavage, then 3 ….

Minor criticism

L 10+l 49  “..is a highly conserved..” remove highly. What would be the difference between a weakly and highly conserved pathway. There is no classification scheme for this. 

L 25  “..the most important…” this is a subjective classification. Please do not use a superlative. 

L 36.  “whereas”,  should be “where”

L 38  “..could not be ..” should be “.. is not produced…”

L 59 “coving” should be “covering”

L 62 “Over 100 years…” should be “More than 100 years…”

L621 “…there are too many…”. Not clear what the author wish to say. Simply write “..there are numerous fascinating…”

I have not listed a few more minor language issues, which may be spotted and corrected a further round of thorough proof reading.

.

Author Response

We thank the editor and the reviewers for critical reading of our manuscript and for their valuable suggestions. In the revised version of manuscript, we have made corrections, removed unnecessary figures and improved the quality of writing. We are confident that we have addressed the reviewers’ concerns in the current form of our manuscript. Our point-by-point response to the reviewers and editorial comments are listed as below and marked in red. Corrections made in the revised manuscript are also marked in red.

Reviewer 2:

The scholarly written review by Chen et al. reviews the literature about notch signaling in insects. Although numerous reviews have been published about Notch signaling in Drosophila, no such review is available that pretty systematically includes studies of Notch signaling in other insects in comparison to what we know from Drosophila. In every chapter of the review, the authors summarize the current understand of Notch signaling and then expand to other insects. Focusing in all chapters on molecular mechanisms, the review nicely complements and importantly extends existing entomology literature and especially entomology books. I would expect that the review will be found useful and informative for both entomologists finding out about molecular mechanisms or Drosophilists moving towards other insects. The list of references is long, and should remain like this, although I have not checked whether it comprehensively covers all related aspects.

Beside a few minor point about language, I have a few suggestions which may improve the manuscript and its impact.

  1. The overall structure of the article is fine with me. Chapter 3 “Notch signalign…” should be split into two “3. Neurogenesis” and “4. Embryonic Development”. The initial molecular experiments and mechanistic understanding of Notch were conducted concerning neurogenesis. Neurogenesis should therefore be visible in the structure of the article and should deserve a chapter on its own. The last chapter should be renamed, since “stress responses” are only a minor part of this chapter. A better title would be something like “adult physiology” or “homeostasis” or the like.

We appreciate the reviewer’s suggestions. The section 3 has been split into two parts, which are “3. Notch signaling in insect embryonic neurogenesis” and “4. Notch signaling in insect embryo segmentation”. We have changed the name of the “stress responses” chapter to “Notch signaling in insect physiological activities”.

  1. Figures. I am not happy with the figures. Thinking which figure I could take for a lecture or a seminar, only figure 1 appears to be suitable. Figure 2 showing an evolutionary tree is currently without any information. The authors may compose a figure in which examples are shown of Notch signaling in species of the various branches. Figure 3 appears superficial. The authors may associate representative and the most striking examples cases to the three mechanisms for both Drosophila and other insects. I would recommend to stick to defined cases to avoid becoming superficial. In order to provide a comprehensive overview a table may be added. In a table it is much easier to provide a complete list. The article will have a much wider impact if accessible and useful/informative figures are included in the article.

We appreciate the reviewer’s suggestions. We agree with the reviewer that Figure 2 and Figure 3 do not provide sufficient useful information. Figure 2 and Figure 3 have been removed. Table 1 is included to summarize the roles of Notch signaling across insect species.

  1. Figure 1: Please add numbers to the figure indicating the steps of the pathway. 1 on the left receptor-ligand binding, then 2 proteolytic cleavage, then 3 ….

Numbers are included in the revised Figure 1 to indicate the signal transduction steps.

Minor criticism

L 10+149  “..is a highly conserved..” remove highly. What would be the difference between a weakly and highly conserved pathway. There is no classification scheme for this.

The word “highly” has been removed in both sentences.

L 25  “..the most important…” this is a subjective classification. Please do not use a superlative.

The word “most” has been removed.

L 36.  “whereas”, should be “where”

The sentence has been revised.

L 38  “..could not be ..” should be “.. is not produced…”

The sentence has been revised.

L 59 “coving” should be “covering”

This typo has been corrected.

L 62 “Over 100 years…” should be “More than 100 years…”

The sentence has been revised.

L621 “…there are too many…”. Not clear what the author wish to say. Simply write “..there are numerous fascinating…”

The sentence has been revised.

I have not listed a few more minor language issues, which may be spotted and corrected a further round of thorough proof reading.

We have thoroughly revised the manuscript to improve the language and modifications are marked in red.